# Influence of Laser Surface Texture on the Anti-Friction Properties of 304 Stainless Steel

Xiashuang Li *, Guifeng Li, Yuesui Lei, Lei Gao, Lin Zhang and Kangkang Yang

College of Mechanical and Electrical Engineering, Shaanxi University of Science and Technology, Xi'an 710021, China
* Correspondence: lixiashuang@sust.edu.cn

**Abstract:** To enhance the anti-friction properties of 304 stainless steel, friction experiments were conducted on it after laser surface texturing. The influences of laser scanning speed, repetition frequency, processing times, laser beam line spacing, and lattice spacing on the friction properties of 304 stainless steel were investigated by contrast tests under annular filling mode. The results revealed that laser texturing improved the anti-friction properties of 304 stainless steel. The friction coefficient of the sample surface decreased first and then increased with the increase in scanning speed, repetition frequency, processing times, laser beam line spacing, and lattice spacing. Based on this, process optimization found that a stainless steel surface with good anti-friction properties could be obtained when the laser power was 0.3 W, the repetition frequency was 50 kHz, the scanning speed was 80 mm/s, the laser beam line spacing was 1 μm, the lattice spacing was 200 μm, and the number of processing times was two. Finally, scanning electron microscope (SEM) characterization of wear morphology on the sample surface showed that the laser textured surface could collect debris during effective friction, which reduced the occurrence of abrasive and adhesive wear. Meanwhile, the actual contact area of the friction pair was effectively reduced, thereby reducing friction force and wear. This study provided experimental data and a theoretical basis for improving the friction properties of the 304 stainless steel surface and laid the foundation for its reliable use under friction and wear conditions.

**Keywords:** 304 stainless steel; laser surface texturing; friction properties

## 1. Introduction

With its excellent corrosion resistance, 304 austenitic stainless steel has been widely employed in many areas of modern industrial production [1,2]. However, its poor wear resistance limits its applications in many engineering fields [3–5]. Surface deformation hardening [6] uses mechanical means, such as shot blasting [7], extrusion [8], roll pressing [9], and laser shock hardening [10,11], to produce strong plastic deformations on metal surfaces in each direction. Therefore, the grains on the surface of the material are refined to form a hardening layer of deformation, increasing the hardness of the surface layer. Furthermore, surface deformation strengthening causes grain breakage and high-density dislocation of metal surfaces. After sufficient time, the plastic flow of metal materials occurs on the workpiece surface. It produces plastic deformation of the overlapping holes, which causes compressive stress and stretches the surface structure during pit generation. Parts inside the workpiece that are not hammered obstruct this process, causing residual compressive stress to accumulate on and close to the workpiece surface, greatly enhancing the mechanical characteristics of the material. However, surface deformation strengthening may induce martensite phase transformation, the varying degrees of which affect the corrosion resistance of austenitic stainless steel differently. The application of surface strengthening in austenitic stainless steel is constrained by this, which is a significant factor. Nitriding as a surface strengthening treatment can enhance the mechanical characteristics

of austenitic stainless steel [12,13]. Traditional gas nitriding requires a high temperature and a long time. Although it improves the wear resistance of surface and the hardness of the material, chromium nitrides are released from thermal decomposition, resulting in a decrease in the corrosion resistance of the material [14–17]. In recent years, various surface texturing methods have emerged, which can be roughly divided into mechanical processing, energy beam etching, and chemical etching. Mechanical processing methods mainly include scanning probe processing. Energy beam etching mostly uses laser and electrochemical etching to process textures. The main chemical etching methods include anodic oxidation and sol-gel [18]. Costa et al. [19] found that circular pit texture with a low area ratio under oil-rich conditions could not improve surface friction performance. However, for strip grooves with a high texture density on the surface (approximately 25%), the friction coefficient could be significantly reduced when the direction of the groove was perpendicular to the direction of motion. However, when the groove direction was parallel to the direction of motion, the friction coefficient increased, and the material exhibited poor wear resistance. Evidently, the texture distribution had an impact on tribological properties. Hua et al. [20] used laser processing technology to fabricate high-precision micro groove arrays on the surfaces of the stator and rotor of the ultrasonic motor. They found that textured surfaces can significantly improve the friction coefficient and lower the wear rate of contact surface, thus increasing the dry friction driving force between the motor stator and rotor. Moreover, the friction coefficient of the double-sided texture is higher than that of the single-sided texture. Xue et al. [21] prepared dimple textures with different area densities on Ti-20Zr-10Nb-4Ta alloy and investigated their tribological properties. It was found that when the texture diameter was fixed with the surface density of the texture from 10% to 40%, the COF and wear rate of the sample surface decreased significantly, and the wear mechanism on the sample surface was abrasive wear and fatigue wear. Siripuram et al. [22] studied the effect of the texture cross-section shape on hydrodynamic lubrication, finding that the friction coefficient did not change much when the depths were constant for microconvex and micropits with various cross-section shapes, but the effect of the cross-section shape on the cross-section area was significant, and both the shape of the cross-section and the size of the texture had significant effects on the leakage rate. Through theoretical investigation, Singh et al. [23] predicted the impact of spherical texture on the performance traits of a two-lobe journal bearing system. They discovered that spherical textures with different dimple aspect ratios can lower the liquid film friction coefficient and enhance bearing performance. Chen et al. [24] fabricated micropore and micrograting arrays on 316 L stainless steel surfaces with a nanosecond laser and tested their friction properties. The results revealed that under the same load and sliding speed, the wear marks on the smooth surfaces were deep and wide, while those on the textured surfaces were shallow and narrow. In addition, the variations in the friction resistance of the textured surface were smaller than those of the smooth surface. The denser the micropore and micrograting arrays, the smaller the friction resistance. Gajrani et al. [25] fabricated mechanical microtextures over the rake face of PN-HSS cutting tools through mechanical microtexturing method. Under the condition of little impact on tool hardness, the performance of the tool is improved by reducing the actual contact area between the tool and chip, and the proper lubrication of contact area through mechanical microstructures. Li et al. [26] used laser microprocessing to form groove textures on the smooth surface of 316 L stainless steel. The results showed that the contact load had little influence on the friction coefficient of the textured surface, while texturing significantly affected friction reduction and anti-wear performance under high loads. Panigrahi et al. [27] investigated the effect of positive deterministic textures with different cross-sectional shapes, sizes, and heights on the tribological properties of parallel sliding lubricated contacts, and found that weaves with fewer texture scales and larger texture heights provided better tribological and lubrication properties to the specimens. When Bharatish et al. [28] used a femtosecond laser to study the effects of scanning speed on the femtosecond laser texture characteristics of a tin bronze alloy containing 8% and 12% tin, they discovered that the wear rate of

the textures produced at lower scanning speeds was lower at higher loads and higher frequencies. Researchers have conducted the studies on the laser preparation technology of micro-circular texture. Deng [29] placed four types of micro-circular textures on the upper surface of rails and performed lubricated reciprocal friction tests. The findings demonstrated that in comparison to the unwoven rail, all four varieties of micro-circular textures enhanced contact stress on the contact surface and decreased friction force during the stable friction phase. Circular cratered micro-circular textures demonstrated the most notable reduction in friction coefficient when compared to linear and lattice-line micro-circular textures. Borjalia et al. [30] found that micro-circular texture reduces material wear by laser ablation of polyethylene materials with different shapes and densities of micro-circular texture. Suitable densities of micro-circular texture structures were obtained by orthogonal experiments. Higher micro-circular texture densities were shown to be ineffective at reducing friction. By adding micro-circular textures to the surface of pure titanium drilling tools, Li [31] investigated the tribological characteristics of these materials when exposed to soil with various particle sizes. The findings demonstrate that when the abrasive's grain size is less than the micro-circular texture's diameter, the abrasive will penetrate the texture and lower the friction coefficient by sliding and rolling. The abrasive will create a sliding plow on the surface of the micro-circular texture when the particle size is greater than the diameter of the texture. While the two abrasives function differently, they can lessen the wear on drilling bits made of pure titanium.

A thorough literature survey reveals that in the above-mentioned research on the influence of surface texture on the properties of metal surfaces, some scholars use mechanical micromachining and photochemical processing to prepare the metal surface texture. When mechanical micromachining is used to prepare metal surface texture, the metal to be processed will suffer relatively large stress, which requires a high degree of hardness of the metal itself; and when using photochemical processing technology to build metal surface texture, the process of preparing the surface texture is more complicated, and it is difficult to control the etching time of the metal when using this method. Other research, meanwhile, using laser surface modification technology to construct texture on the metal surface, mainly focus on the study of the influence of texture shape, area density, and size on various characteristics of the metal material surface. In such studies, the laser processing parameters used are fixed and the types of metals studied vary with the properties of the metals affected by the surface weaving. There is little research on the tribological properties of 304 stainless steel by using laser surface texture technology to construct texture on the surface of 304 stainless steel, and there were still some deficiencies in the studies on laser parameters affecting the friction properties of material surface by affecting texture. The friction mechanism of 304 stainless steel surface after laser surface texture needs further research.

This article presents a thorough review of literature and focuses on the study of 304 stainless steel based on previous research. A nanosecond laser was used in an annular filling mode to investigate the impact of various laser parameters (such as scanning speed, repetition rate, processing times, and laser beam line spacing) on the surface friction performance of 304 stainless steel. The study also used a single-factor rotation method to evaluate the effect of surface texture created by different texture lattice spacing on surface friction performance. The surface morphology and depth of the texture were analyzed using an ultra-deep field microscope. Additionally, friction experiments were conducted on the surface of the 304 stainless steel before and after texturing using a friction and wear tester to examine the differences in surface friction performance. The surface of the textured 304 stainless steel was observed using a scanning electron microscope after friction experiments, and the friction mechanism of 304 stainless steel surface after laser surface texturing was further studied. The goal of this study is to enhance the anti-friction performance of 304 stainless steel surface and to provide a comprehensive and systematic laser processing solution.

## 2. Experiment

### 2.1. Testing Apparatus and Materials

The maximum laser power of the fiber laser marker used in the experiment was 20 W. Austenitic 304 stainless steel sheets were selected and cut by laser into round 40 mm samples with a thickness of 6 mm. The chemical composition is shown in Table 1.

**Table 1.** Chemical composition of 304 stainless steel austenitic [32].

| Element | C | Si | Mn | Cr | Ni | S | P | Fe |
|---|---|---|---|---|---|---|---|---|
| Proportion (%) | 0.07 | 1.00 | 2.00 | 17.15 | 8.02 | 0.03 | 0.035 | 71.965 |

### 2.2. Experiment Design

Using a nanosecond fiber laser marking machine (HGTECH-LG20, Wuhan Huagong Laser) with an emission wavelength of 1064 nm, a maximum average power of 20 W, a pulse duration of 200 ns, and a repetition frequency that can be continuously adjusted within the range of 30 kHz to 60 kHz. The laser beam was focused onto the substrate by means of a 110 mm focusing lens, resulting in a laser spot diameter of 50 μm after focusing. The machine was equipped with a galvanometer scanner, which was responsible for completing the texturing process.

The surface texture of the sample is processed by laser using the scanning strategy shown in Figure 1. Figure 1a shows the theoretical processing method, where overlapping of laser spots occurs when the distance between laser beam lines is smaller than the diameter of the laser spot, otherwise they will not. Under high magnification, some of the textures produced by the processing appear as polygons. This is because the galvanometer scanning system control software employs an annular filling method, as shown in Figure 1b. As a result, the laser path is automatically fitted from polygons to circles when processing the texture.

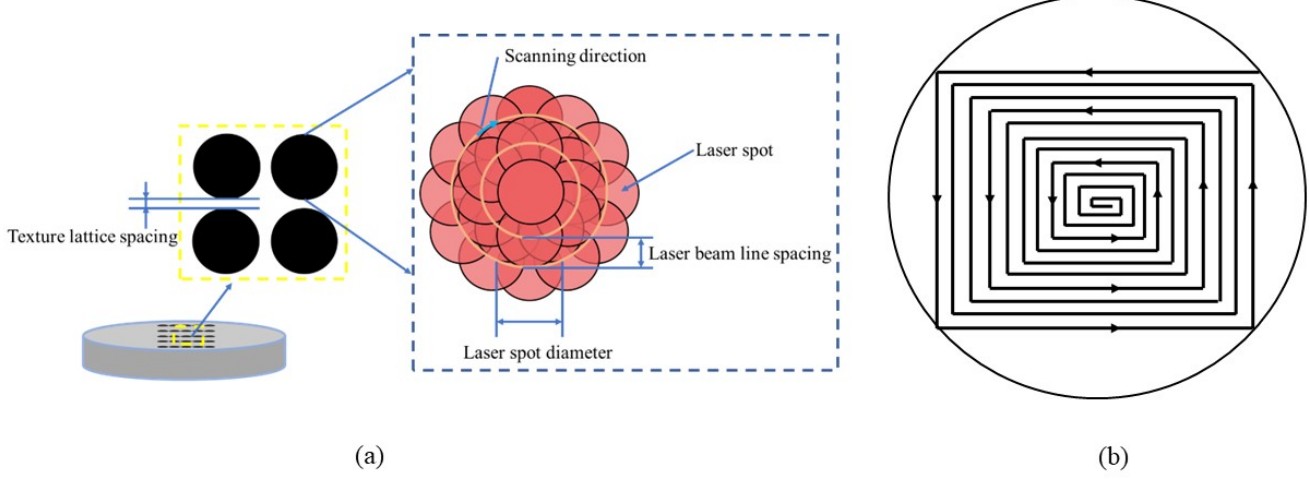

(a)　　　　　　　　　　　　　　　　　　　　　　　　　　　　　(b)

**Figure 1.** Scanning strategy for surface texture produced by laser: (**a**) theoretically, (**b**) actually.

During laser processing, a significant amount of slag remains on both the surface and inside the texture of the sample, affecting its roughness. These slags can have a profound impact on the depth of the texture's subsequent characterization. Therefore, using 2000# sandpaper to lightly polish the sample can remove the slag remaining on the surface of the sample without damaging the texture of the laser processed surface. Additionally, prior to conducting friction tests, it is essential to use ultrasonic cleaning to eliminate any slag or impurities inside the texture. This process not only enhances the texture's ability to collect debris but also prevents any polishing dust from entering the texture and affecting

subsequent experiments. Moreover, ultrasonic cleaning makes the depth and morphology of the processed texture clearer.

To investigate the effect of surface texture on the friction properties of 304 stainless steel, the dry friction test was conducted on textured samples. The tests were performed using an electrochemical corrosion friction tester (MSR-2T) and a GGr15 steel ball with a diameter of 6 mm as the friction sub-mate material, as this type of steel is commonly used in practical applications with 304 stainless steel. The samples were fixed on a special fixture on the chassis, while the GGr15 steel ball was fixed on the upper end of the machine fixture. A loading force perpendicular to the sample surface was applied, and the friction test was carried out using a reciprocating linear motion with a 5 mm stroke and a 5 N applied load for 30 min. The surface morphology and internal structure of the substrates were examined using an ultra-deep field microscope (DSX-510) and a scanning electron microscope (SU3500), while the hardness of the samples before and after laser processing was measured using a digital micro-Vickers hardness tester (402SXV). For more information on the test process, please see Table 2.

**Table 2.** Test scheme for laser surface texturing.

| No. | Laser Power/W | Scanning Speed/mm/s | Repetition Frequency/kHz | Processing Times | Laser Beam Line Spacing/μm | Lattice Spacing/μm |
|---|---|---|---|---|---|---|
| 1 | 0.3 | 60 | 60 | 1 | 1 | 200 |
| 2 | 0.3 | 80 | 60 | 1 | 1 | 200 |
| 3 | 0.3 | 100 | 60 | 1 | 1 | 200 |
| 4 | 0.3 | 120 | 60 | 1 | 1 | 200 |
| 5 | 0.3 | 140 | 60 | 1 | 1 | 200 |
| 6 | 0.3 | 100 | 35 | 1 | 1 | 200 |
| 7 | 0.3 | 100 | 40 | 1 | 1 | 200 |
| 8 | 0.3 | 100 | 45 | 1 | 1 | 200 |
| 9 | 0.3 | 100 | 50 | 1 | 1 | 200 |
| 10 | 0.3 | 100 | 55 | 1 | 1 | 200 |
| 11 | 0.3 | 100 | 60 | 2 | 1 | 200 |
| 12 | 0.3 | 100 | 60 | 3 | 1 | 200 |
| 13 | 0.3 | 100 | 60 | 1 | 5 | 200 |
| 14 | 0.3 | 100 | 60 | 1 | 10 | 200 |
| 15 | 0.3 | 100 | 60 | 1 | 50 | 200 |
| 16 | 0.3 | 100 | 60 | 1 | 100 | 200 |
| 17 | 0.3 | 100 | 60 | 1 | 1 | 100 |
| 18 | 0.3 | 100 | 60 | 1 | 1 | 300 |

## 3. Results and Discussions

### 3.1. Effect of Scanning Speed on Surface Texturing and Anti-Friction Properties of 304 Stainless Steel

According to the experiment scheme, groups 1–5 used different scanning speeds to texture the samples. The texture morphologies observed under a super depth-of-field microscope are shown in Figure 2a–e. According to the analysis, as the scanning speed increased, the duration of energy acting at a certain point decreased, resulting in less material removal at that point and a lower average texture depth. This was consistent with the trend of the average texture depths measured experimentally (Figure 2f), showing that as the scanning speed increased, the average texture depth decreased. The trend of the friction coefficient of the material surface over time at various scanning speeds is shown in Figure 2g. The friction coefficient was very high at the beginning of the friction process. This is because at the beginning of grinding, there is still a small amount of residual molted material on the surface, resulting in excessive surface roughness. After 20 min, the friction coefficient gradually decreased and stabilized in the stable wear stage. This is because the texture acts as a trap for impurities such as abrasive dust, reducing the surface roughness of the substrate material while reducing abrasive wear on the surface of the substrate material. As displayed in Figure 2h, when the scanning speed reached 80 mm/s, the friction coefficient was the lowest at 0.5692. However, when the scanning speed increased further to 140 mm/s, the friction coefficient increased. A preliminary analysis indicated

that the reason for this phenomenon was that when the scanning speed was too high, the time for the laser to act on one specific point on the material surface became shorter and there was poor continuity of the laser spot. Energy absorption was insufficient for effective removal of surface material. Instead, an oxidized layer was produced that could be peeled off more easily and has a higher roughness than the matrix material on the surface, resulting in a further increase in the friction coefficient. Therefore, within the appropriate scanning speed range, the friction coefficient of the laser-textured surface was lower than those of the nontextured surface, which proved that laser surface texturing could improve the anti-friction properties of materials. According to the analysis, deep textures on the material surface could mitigate abrasive and adhesive wear by collecting debris generated during the friction process, thus improving the anti-friction properties of the material surface. In other words, when the scanning speed was 80 mm/s, the prepared surface texture resulted in the lowest friction coefficient and the best anti-friction properties.

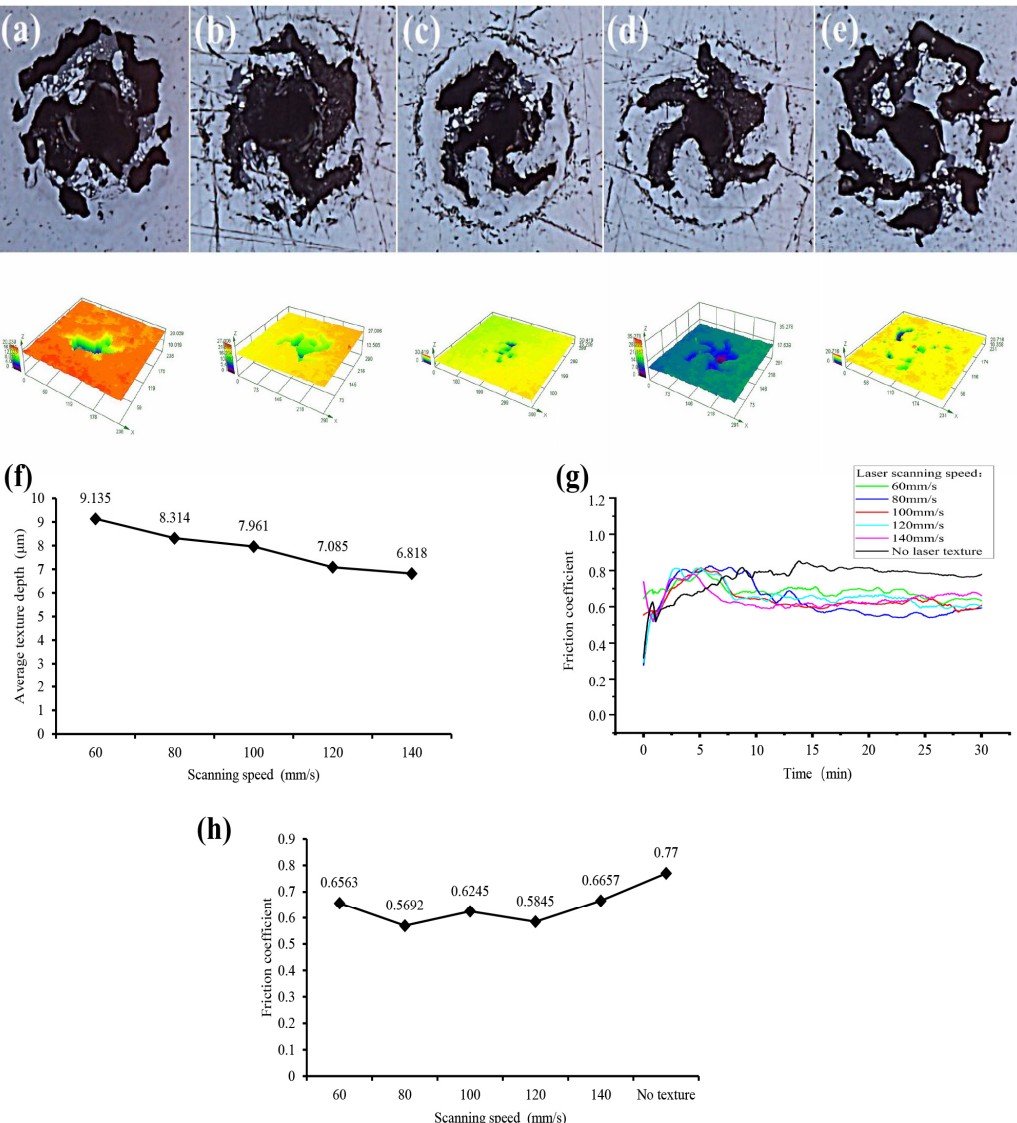

**Figure 2.** Texture morphology of sample surface at different scanning speeds. (**a**) 60 mm/s, (**b**) 80 mm/s, (**c**) 100 mm/s, (**d**) 120 mm/s, (**e**) 140 mm/s, (**f**) average depth of textures processed at different scanning speeds, (**g**) variation of friction coefficient with time at different scanning speeds, and (**h**) the friction coefficient of stable wear at different scanning speeds.

### 3.2. Effect of Repetition Frequency on Surface Texturing and Anti-Friction Properties of 304 Stainless Steel

Based on the first groups of investigations, groups 6–10 used a different repetition frequency to texture the samples. By observing the texture morphologies under the super depth of the field microscope as displayed in Figure 3a–f, it was found that with the increase in repetition frequency, the energy density acting at a certain point decreased, resulting in less material removal and a smaller texture depth. This was consistent with the average texture depths experimentally measured (Figure 3g). The overall average texture depth decreased with increasing repetition frequency. The trend of the friction coefficient for the material surface over time at various repetition frequencies is shown in Figure 3h, showing a very high friction coefficient at the beginning of the wear stage. This is because at the beginning of grinding, there is still a small amount of residual molted material on the surface, resulting in excessive surface roughness. After 20 min, the friction coefficient gradually decreased and stabilized when entering the stable wear stage. This is because the texture acts as a trap for impurities such as abrasive dust, reducing the surface roughness of the substrate material while reducing abrasive wear on the surface of the substrate material. As show in Figure 3i, when the repetition frequency was 50 kHz, the friction coefficient was 0.5273, which was smaller than those at other repetition frequencies. In the stable wear stage, with increasing repetition frequency, the friction coefficient gradually increased, the energy acting on the surface of the material at a single time decreased, and the texture depth decreased. However, the friction coefficient of the surface of the textured material was smaller than that of the nontextured material. The analysis revealed that the anti-friction properties of the material surface improved, which was consistent with the fact that the friction coefficient was smaller. Therefore, when the repetition frequency was 50 kHz, the prepared surface texture resulted in the lowest friction coefficient and the best anti-friction properties.

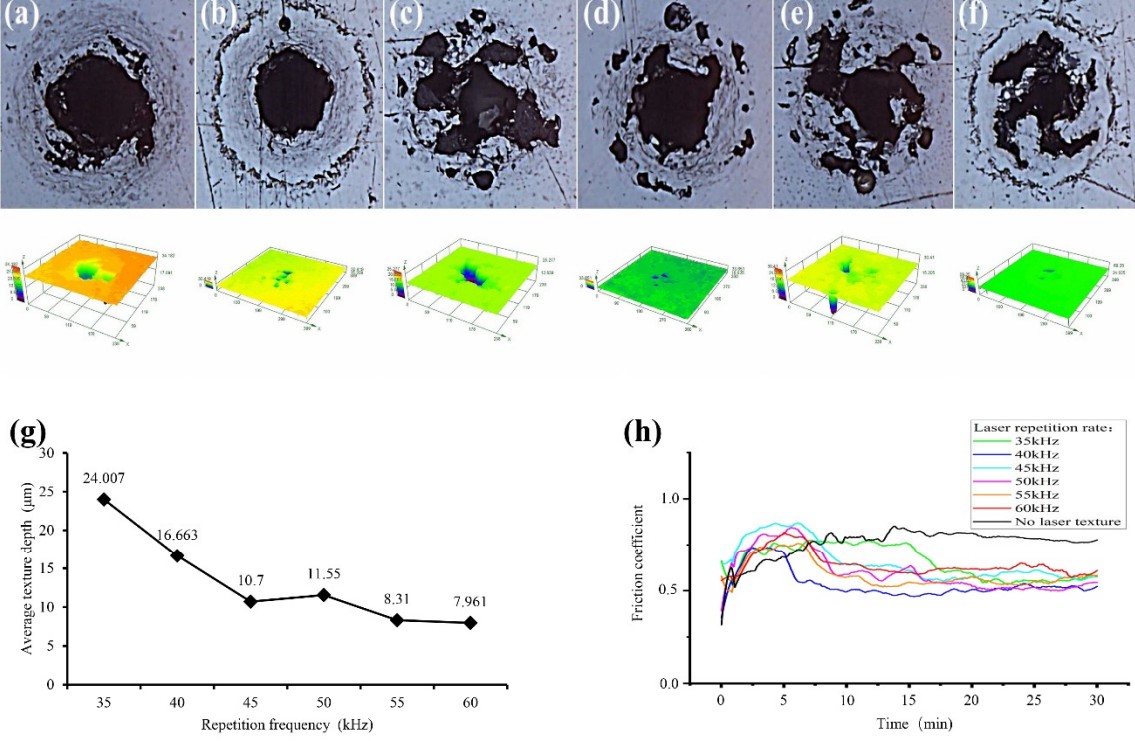

**Figure 3.** *Cont.*

**(i)**

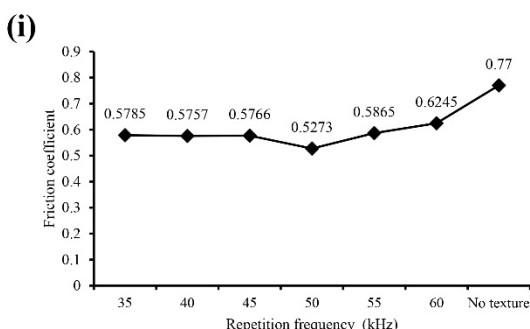

**Figure 3.** Texture morphology of sample surface at different repetition rates. (**a**) 35 kHz, (**b**) 40 kHz, (**c**) 45 kHz, (**d**) 50 kHz, (**e**) 55 kHz, (**f**) 60 kHz, (**g**) average depth of textures prepared at different repetition rates, (**h**) variation of friction coefficient with time at different repetition rates, (**i**) the friction coefficient of stable wear at different repetition rates.

### 3.3. Effect of Processing Times on Surface Texturing and Anti-Friction Properties of 304 Stainless Steel

Based on the first groups of investigations, groups 11–12 used different processing times to texture the samples. By observing the texture morphologies under the super depth of field microscope, as displayed in Figure 4a–c, it was found that as the processing times increased, the sum of laser energy acting on a certain point increased, the amount of removed material increased, and the average texture depth increased as well. This was consistent with the average texture depths experimentally measured and displayed in Figure 4d. The average texture depth increased with the number of processing times. The trend of the friction coefficient of the material surface over time at various processing times is shown in Figure 4e, showing that the friction coefficient was very high at the initial wear stage. This is because at the beginning of grinding, there is still a small amount of residual molted material on the surface, resulting in excessive surface roughness. After 15 min, the friction coefficient gradually decreased and stabilized when entering the stable wear stage. This is because the texture acts as a trap for impurities such as abrasive dust, reducing the surface roughness of the substrate material while reducing abrasive wear on the surface of the substrate material. As shown in Figure 4f, the friction coefficient measured during the stable wear stage was the smallest (0.5865) when processed twice. Increasing the amount of processing times could increase the texture depth and decrease the friction coefficient, but too many processing times could increase the thickness of the recast layer, increasing the friction coefficient. Therefore, when the processing times were 2, the texture of the prepared surface minimized the friction coefficient of the material surface and optimized the anti-friction properties of the material.

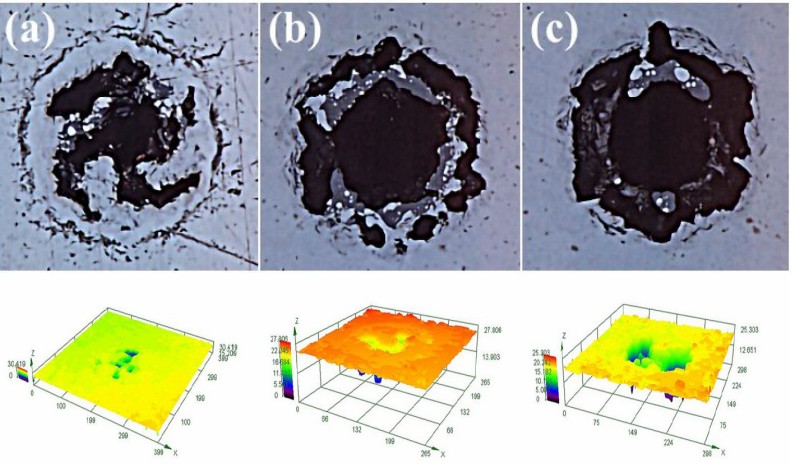

**Figure 4.** *Cont.*

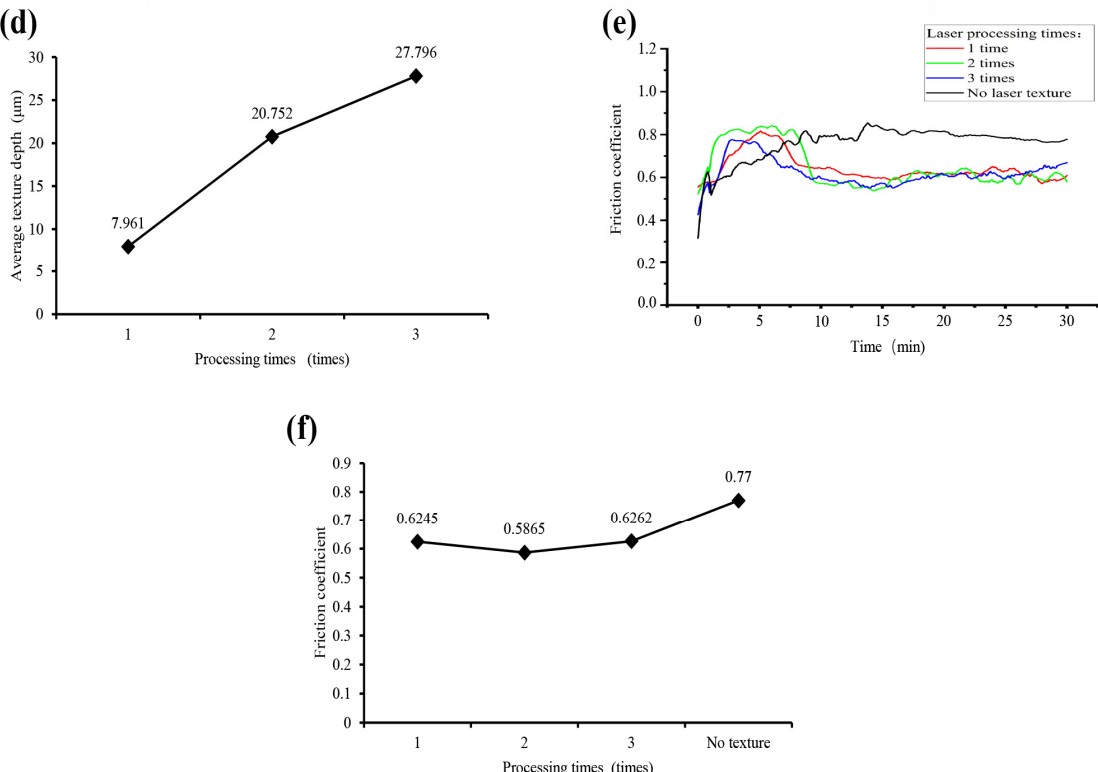

**Figure 4.** Texture morphology of sample surface under different processing times. (**a**) 1, (**b**) 2, (**c**) 3, (**d**) average depth of texture prepared by different processing times, (**e**) variation of friction coefficient with time under different machining times, (**f**) friction coefficient of stable wear under different machining times.

*3.4. Effect of Laser Beam Line Spacing on Surface Texturing and Anti-Friction Properties of 304 Stainless Steel*

Based on the first groups of investigations, groups 13–16 adopted different laser beam line spacing to texture the samples. By observing the texture morphologies under the super-depth of field microscope (Figure 5a–e), it was found that the overlapping area of the laser beam spot decreased with the increase in the laser beam line spacing. This resulted in a smaller energy density on the material surface, less material removal, and a smaller average texture depth. This was consistent with what is shown in Figure 5f, where the average depth of the texture decreased with the increasing spacing of the laser beam lines. The trend of the friction coefficient of the material surface over time under various laser beam line spacing conditions is shown in Figure 5g, which shows a very high coefficient of friction at the beginning of the wear stage. This is because at the beginning of grinding, there is still a small amount of residual molted material on the surface, resulting in excessive surface roughness. After 15 min, the friction coefficient gradually decreased and stabilized when entering the stable wear stage. This is because the texture acts as a trap for impurities such as abrasive dust, reducing the surface roughness of the substrate material while reducing abrasive wear on the surface of the substrate material. When the line spacing was much smaller than the laser spot diameter, a large energy density acted at the same point on the material surface, causing a large amount of material removal and deeper textures, which could effectively improve the anti-friction properties of the material by collecting debris. The friction coefficients measured during the stable wear stage (Figure 5h) showed that when the laser beam line spacing was 1 μm, the friction coefficient was the smallest at 0.6245. The friction coefficients of laser-textured material surfaces were smaller than those of the nontextured ones. However, when the line spacing was 5 μm, the friction coefficient of the material surface was the highest. Different line spacings resulted in varied texture morphologies. In Figure 5d,e, a line spacing larger than the minimum laser spot diameter

was used, so the laser spots did not overlap on the processing path and materials were only removed on the laser path. The friction coefficient was taken as the main parameter to evaluate the friction properties of samples. When the laser beam line spacing was 1 µm, the texture of the prepared surface minimized the friction coefficient of the material surface.

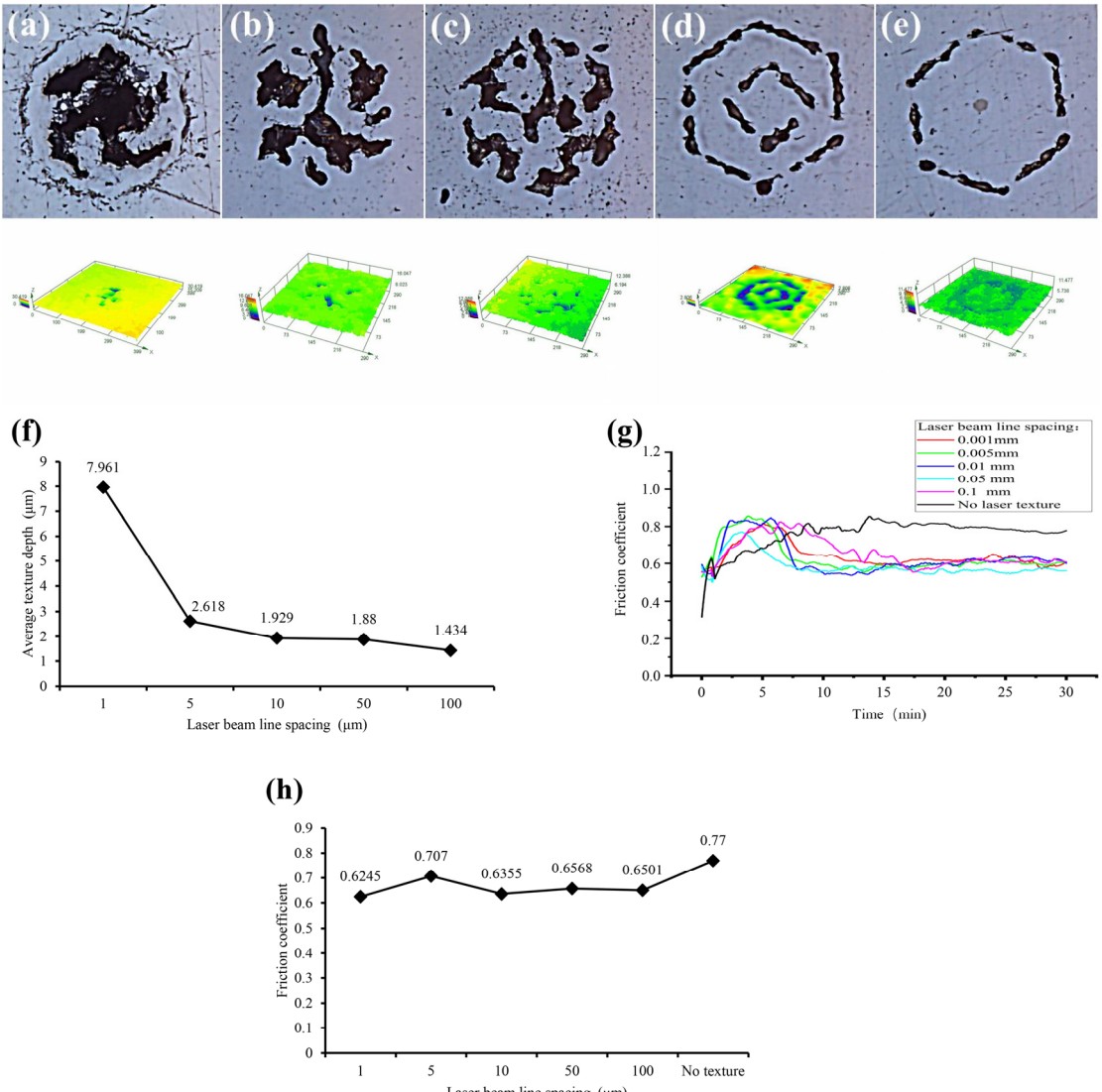

**Figure 5.** Texture morphology of sample surface under different line spacing. (**a**) 1 µm, (**b**) 5 µm, (**c**) 10 µm, (**d**) 50 µm, (**e**) 100 µm, (**f**) average depth of textures prepared with different line spacing, (**g**) change of friction coefficient with time under different laser beam line spacing conditions, (**h**) friction coefficient of stable wear at different line spacing.

### 3.5. Effect of Texture Lattice Spacing on Surface Texturing and Anti-Friction Properties of 304 Stainless Steel

Based on the first groups of investigations, groups 17–18 used different lattice spacing to texture the samples. The morphologies of the textures observed under the super depth of the field microscope are shown in Figure 6a–c, and the average depths of textures experimentally measured are displayed in Figure 6d. According to the analysis, since the laser parameters did not change, the textures under various lattice spacings (i.e., texture densities) exhibited little difference in texture morphology and average texture depth. The trend of the friction coefficient of the material surface over time under various lattice spacing conditions is shown in Figure 6e, showing a very high friction coefficient in the initial wear stage. This is because at the beginning of grinding, there is still a small amount

of residual molted material on the surface, resulting in excessive surface roughness. After 20 min, the friction coefficient gradually decreased and stabilized when entering the stable wear stage. This is because the texture acts as a trap for impurities such as abrasive dust, reducing the surface roughness of the substrate material while reducing abrasive wear on the surface of the substrate material. The stable wear stage that the friction coefficient was the lowest at was 0.6245, as reflected in Figure 6f, when the lattice spacing was 200 μm. The texture density varied with the lattice spacing, but a suitable range existed. When the lattice spacing was moderate, the textures exhibited the best debris collection capability and the greatest improvement in the anti-friction properties of the material. The friction coefficient was taken as the main parameter to evaluate the friction properties of the samples. When the lattice spacing was 200 μm, the prepared surface texture made the friction properties of the material surface optimal.

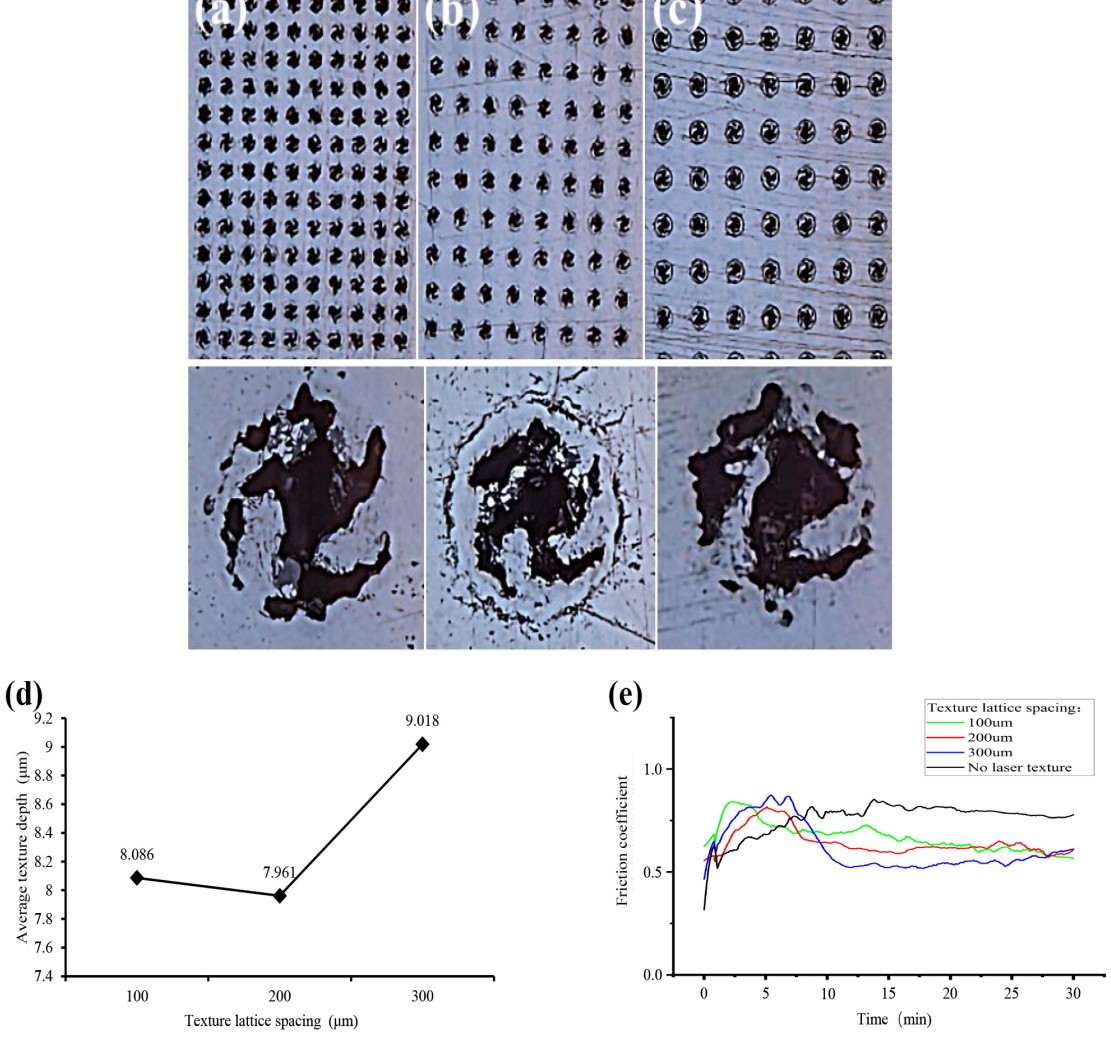

**Figure 6.** *Cont.*

**(f)**

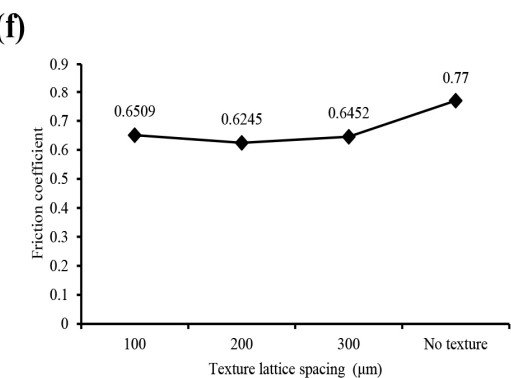

**Figure 6.** Surface morphology of samples with different textural lattice spacing. (**a**) 100 μm, (**b**) 200 μm, (**c**) 300 μm, (**d**) the average texture depth was prepared with different lattice spacing, (**e**) variation of friction coefficient with time under different lattice spacing, (**f**) friction coefficient of stable wear at different lattice spacing.

### *3.6. Effect of Surface Texture on Material Surface Hardness*

In this test, the hardness of the material surface before and after processing the texture was measured and compared using a micro-Vickers hardness tester. The hardness of the material surface before processing the texture was 228.6 HV using the method of multiple measurements at multiple locations and taking the average value, and the hardness of the material surface after processing the texture was about 228.6 HV. Using the same method, the hardness of the different texture surfaces processed under various laser parameters was measured at around 229.70 HV, with a maximum value of 245.73 HV and a minimum value of 213.80 HV, as shown in Figure 7. The hardness of the material surface did not change significantly before and after the processing of the weave, indicating that the weave studied in this test did not cause significant changes in the hardness of the material, which in turn did not affect the frictional wear performance of the sample.

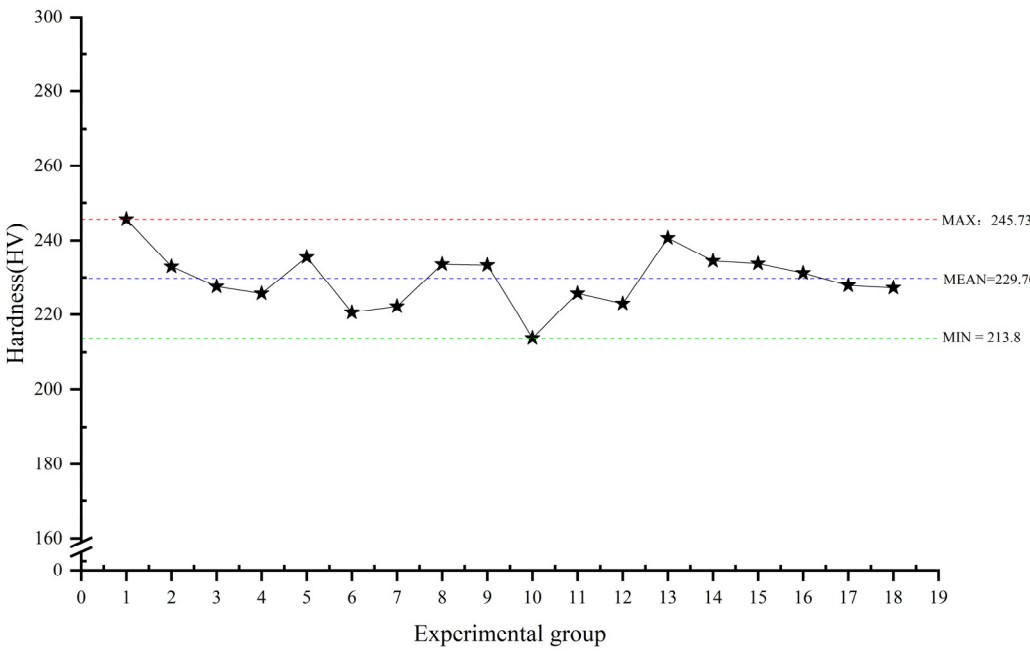

**Figure 7.** Average surface hardness of each test group after texturing treatment.

### **4. Wear Mechanism Analysis**

The micromorphologies of the abrasion marks under various surface textures are displayed in Figure 8. Figure 8a shows the abrasion marks on the surface before laser

texture. Figure 8b is the result after processing once with a laser power of 0.3 W, a repetition frequency of 60 kHz, a scanning speed of 100 mm/s, a laser beam line spacing of 1 μm, and a lattice spacing of 200 μm. The other abrasion marks in Figure 8 were obtained after changing a single variable according to Figure 8b. The abrasion morphologies on the textured surfaces processed under various parameters were similar to those without texturing, indicating that their wear mechanisms were the same. There were varying numbers of furrows, pits, and scaly fragments in each abrasion morphology in Figure 8. The debris generated during the wear process would become abrasive grains and produce clear grooves on the friction surface, called furrows, which was typical for abrasive wear. During the wear process, the friction pair moved relative to each other and their surfaces were subjected to cyclic loads, which were prone to deformations and cracks. As the cracks expanded, the materials would be peeled off to form pits, which was a typical characteristic of fatigue wear. As the friction progressed, the temperature of the friction surface increased and the wear debris adhered to the surface of the material, forming scaly fragments, which then caused the adhesive to wear [33].

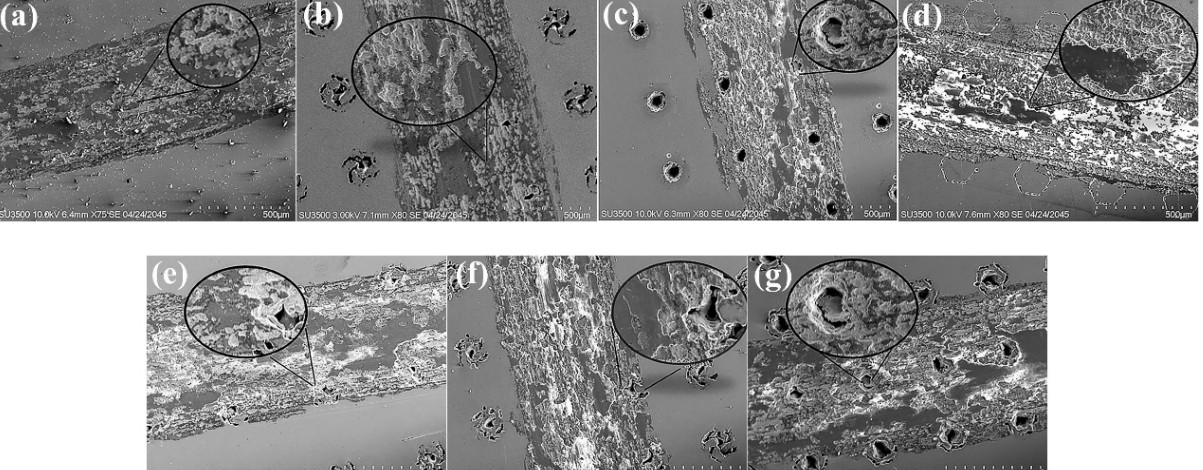

**Figure 8.** Micromorphologies of abrasion marks under surface texture. (**a**) Nontextured state, (**b**) Laser power 0.3 W, repetition frequency 60 kHz, scanning speed 100 mm/s, processed once, laser beam line spacing 1 μm, lattice spacing 200 μm, (**c**) repetition frequency 40 kHz, (**d**) laser beam line spacing 100 μm, (**e**) lattice spacing 300 μm, (**f**) scanning speed 120 mm/s, (**g**) processed twice.

Observation of each micromorphology revealed that the furrows on both sides of the abrasion mark were more obvious, whereas pits and scaly fragments were more frequent in the middle part. Analysis showed that abrasive wear occurred first as the main wear mode during the wear process and debris was generated during the movement of the friction pair. On the one hand, furrows were generated on the material surface; on the other hand, due to the increasing temperature, the debris adhered to the material surface, turning the type of wear to adhesive wear. During the whole process, cracks occurred on the material surface due to cyclic loading, which led to the material peeling and belonged to the fatigue wear stage. The peeled materials were then involved in abrasive and adhesive wear, such as abrasive grains and debris.

As shown in Figure 8a, adhesive wear and fatigue wear were the main types of wear for the nontextured surface. In Figure 8b,d,e, the overall depths of the textures were small, the texture surfaces were not very large, and there were more scaly fragments than pits, indicating that the adhesion wear was dominating. Figure 8c,f,g showed that the pits were in the majority, indicative of fatigue wear. Figure 8c,e show that the textures collected some debris. In summary, the added surface textures could collect debris during friction, thereby reducing the occurrence of abrasive and adhesive wear and thus the degree of wear.

Bowden proposed adhesive friction theory [34], revealing that when adhesive wear occurs, the relative sliding of the bond between the two friction surfaces will destroy the bonding point and produce a shear force, called the friction force. In this experiment, hard bearing steel and soft 304 stainless steel were the two surfaces subjected to friction, for which there was a hardness difference. Therefore, furrows formed on the surface of stainless steel with a smaller hardness. The force that produced the furrows was also part of the friction force.

Thus, in this experiment, the surface textures reduced friction and wear by lowering the actual contact area between the two friction surfaces.

## 5. Conclusions

In this work, a nanosecond laser marking machine was used to process the anti-friction textured surface of 304 stainless steel. The research results are summarized as follows:

(1) The selected texture filling method for this experiment is annular filling, which involves higher energy density more at the corners than in the straight-line regions. This leads to more melting and vaporization of material in the corner area, causing material to splash onto the surrounding region and accumulate. After undergoing slight polishing and ultrasonic cleaning, the laser processed texture displays deeper texture depth at the corners with annular filling. This results in a petal-shaped depression instead of the circular depression that was designed in theory.

(2) Hardness measurements have shown that there was no significant difference in the material's hardness on the surface of 304 stainless steel before and after laser processing was used to create a texture on it. This indicates that the surface texture does not cause any changes to the surface hardness of the material.

(3) Optimizing laser parameters can enhance the anti-friction performance of the 304 stainless steel surface by modifying the texture morphology. The experimental results indicate that decreasing the scanning speed, repetition frequency, and laser beam line spacing while maintaining an appropriate lattice spacing leads to a gradual increase in texture depth and better debris collection ability. The size and morphology of the structure are influenced by laser parameters. Thus, it can be concluded that using laser power of 0.3 W, repetition frequency of 50 kHz, scanning speed of 80 mm/s, laser beam line spacing of 1 μm, lattice spacing of 200 μm, and processing times of two yields per texture with distinct contours and efficient debris collection. This, in turn, reduces abrasive and adhesive wear.

(4) Through observation of surface scratches on the sample and analysis of the wear mechanism, it was discovered that in the absence of laser surface texturing, adhesive wear, abrasive wear, and fatigue wear all occur simultaneously. On the other hand, textured surfaces primarily experience fatigue wear. The designed texture can effectively collect the wear debris generated during the wear process, reducing the incidence of abrasive wear and adhesive wear. This, in turn, improves the material's frictional performance on the surface.

**Author Contributions:** X.L.: Conceptualization, Supervision, Methodology, Writing–original draft. G.L.: Investigation. Y.L.: Investigation. L.G.: Investigation. L.Z.: Investigation. K.Y.: Investigation. All authors have read and agreed to the published version of the manuscript.

**Funding:** The author(s) disclosed receipt of the following financial support for the research, authorship, and publication of this article: This work was supported by China Postdoctoral Science Foundation (2020M683406), Natural Science Foundation of Shaanxi province (2018JQ5201) and Science and technology Project of Education Department of Shaanxi Province (21JK0538).

**Acknowledgments:** The authors thank the members of our laboratory for their aid in sample preparation and analysis during the tests. The authors would also like to thank He Nairu and Chen Wei, members of the Center for New Technology and Service Behavior of Mechanical Surfaces in Shaanxi University of Science and Technology for their help in the analysis of friction and wear mechanisms.

**Conflicts of Interest:** The authors declare that they are not aware of competing economic interest or personal relationships that may affect the work reported in this paper.

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
