# Peer review of "Influence of Laser Surface Texture on the Anti-Friction Properties of 304 Stainless Steel"

_machines, doi:10.3390/machines11040473_

Round 1
Reviewer 1 Report
In the introduction there are more inaccuracies. Formation of compressive stress and stretches is not explained. "Parts inside the workpiece that are not hammered obstruct...." There is necessary to simply describe the mechamism and effect of surface strengthening.
The introduction, despite being extensive, does not serve the purpose.
Experimment design is not simply described. Authors use very long sentences.
What does it mean "internal structure" ? row 201, page 5
What is diffrence between laser frequency and repetition frequency?
At the beginning of grinding (row 259, page 7) ...residual "molten" material. Is it real molten material? I suppose, that it was molted material. Similar explanation is also in a later text.
Figure 4: (f) "and".... without and
In Figure 3: frequency kHz, not KHz.
Page 9, row 299 - what is "recast layer?"
Figure 6, (c): without :
Page 15, rows 428-430 ???
Conclusions should be better explained.
Reviewer 2 Report
Dear Authors,
Thank you for submitting your work to Machines Journal. Enclosed you find my comments.

Round 2
Reviewer 1 Report
page 2, row 65-66 surface .,
Reviewer 2 Report
Dear Authors,
Thank you for submitting a revised version of your manuscript to Machines Journal.